# A Review of EMI Research of High Power Density Motor Drive Systems for Electric Actuator

Zhenyu Wang [1], Dong Jiang [1,*], Zicheng Liu [1], Xuan Zhao [1], Guang Yang [2] and Hongyang Liu [2]

1 School of Electrical and Electronic Engineering, Huazhong University of Science and Technology, Wuhan 430074, China; wang_zhenyu@hust.edu.cn (Z.W.); liuzc@hust.edu.cn (Z.L.); zhaoxuan95@hust.edu.cn (X.Z.)
2 Flight Automatic Control Research Institute, Xi'an 710076, China; yangg073@avic.com (G.Y.); liuhy076@avic.com (H.L.)
* Correspondence: jiangd@hust.edu.cn; Tel.: +86-186-2792-2372

**Abstract:** With the global attention given to energy issues, the electrification of aviation and the development of more electric aircraft (MEA) have become important trends in the modern aviation industry. The electric actuator plays multiple roles in aircraft such as flight control, making it a crucial technology for MEA. Given the limited space available inside an aircraft, the power density of electric actuators has become a critical design factor. However, the pursuit of high power density results in the need for larger rated power and higher switching frequency, which can lead to severe electromagnetic interference (EMI) issues. This, in turn, poses significant challenges to the overall reliability of the electric actuator. This paper provides a comprehensive review of EMI in high power density motor drive systems for electric actuator systems. Firstly, the state of the art of electric actuator systems are surveyed, pointing out the contradictory relationship between high power density and EMI. Subsequently, various EMI modeling approaches of motor control systems are reviewed. Additionally, the main EMI suppression methods are summarized. Active EMI mitigation methods are emphasized in this paper due to their advantages of higher power density compared with passive EMI filters. Finally, the paper concludes by summarizing the EMI research in motor drive systems and offering the prospects of electric actuators.

**Keywords:** more electric aircraft (MEA); electric actuator; high power density motor drive; electromagnetic interference (EMI); EMI suppression methods

## 1. Introduction

During the 21st century, with the increasing energy shortage crisis and the growing global climate issues, the United Nations has proposed taking urgent action to achieve sustainable development for human society [1,2]. The aviation industry, as an important mode of transportation and a crucial component of the global economy, also faces challenges in achieving sustainable development. In 2018, the aviation industry accounted for approximately 2.4% of $CO_2$ emissions. As air passenger traffic continues to grow at a rate of 4% to 5% annually, the issue of carbon emissions will become even more serious [3]. In response to the energy supply challenges and environmental issues, the European Union's Flightpath 2050 goals aim to reduce $CO_2$ emissions by 75%, $NO_x$ emissions by 80%, and perceived noise emissions by 65% compared to the 2000 baseline [4,5]. Aviation electrification and the rise of more electric aircraft (MEA) have emerged as prominent solutions and are currently dominating the industry.

The concept of MEA has been around for a long time [6]. In recent decades, the rapid development of key technologies such as power electronics conversion, advanced motors, electrochemical energy storage, and high-temperature superconductivity has led to the rapid advancement of electrification [5,7]. The energy management architecture of traditional aircraft is complex, using jet fuel as the primary power for propulsion.

The remainder is then converted into four types of secondary power: pneumatic power, mechanical power, hydraulic power, and electrical power [8]. Meanwhile, a MEA mainly relies on electricity as the secondary power, from generators or batteries. Taking the Boeing 787 as an example [9], the electrical system has replaced most of the pneumatic system, eliminating the traditional bleed manifold. The electric compressors replace the traditional pneumatic systems of engines (expected to reduce energy loss by 35%), and the air conditioning packs and wing anti-icing systems are both driven by electric power. Overall, aerospace electrification can reduce the aircraft's reliance on fossil fuels, thereby reducing the size and weight of the aircraft [10]. However, complex electric power systems will place higher requirements on reliability and system control strategies [11].

With the advancement of MEA, the electric actuator, also called a power-by-wire (PBW) actuator, is gradually replacing the traditional hydraulic actuator [12,13]. As shown in Figure 1, the actuation systems perform multiple functions, including flight controls, landing gear controls, and engine actuation controls [14]. Compared to hydraulic actuators, electric actuators have advantages of a smaller size and weight, as well as higher efficiency. The electric actuators are primarily classified into two types: electro-hydraulic actuators (EHAs) and electro-mechanical actuators (EMAs), as shown in Figure 2. The EHA implements power control by adjusting the flow rate and power of a fix displacement pump using a variable speed motor. This eliminates the need for extensive piping systems and external hydraulic devices, resulting in a more compact design. The EMA implements power control by utilizing the motor and gearbox to directly move a ball screw. Compared to the EHA, the EMA offers evident advantages in terms of weight and efficiency as it does not require hydraulic devices. However, it is commonly believed that EMAs may not offer the same level of reliability as EHAs under the same power level [15]. Therefore, the EMA is more readily accepted for secondary flight controls that have lower power and safety requirements, such as flaps, slats, spoilers, and horizontal stabilizing surfaces. On the other hand, the primary flight controls mainly rely on EHAs.

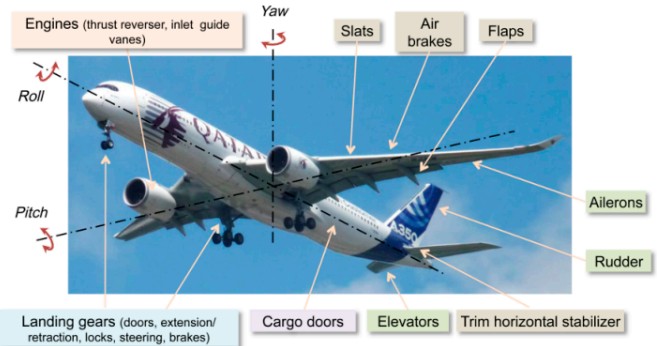

**Figure 1.** Different actuation needs on a commercial aircraft [14].

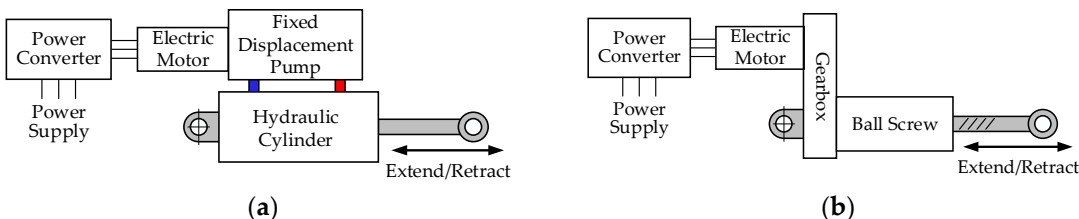

**Figure 2.** The schematic diagrams of EHA and EMA: (**a**) EHA; (**b**) EMA.

The reliability, weight, power density, and efficiency are crucial design indicators for electric actuators. On the one hand, electric actuators are evolving towards distributed design and redundant fault-tolerant design to enhance reliability [16]. As a result, open-winding, multi-phase, and multi-three-phase motors are gaining increasing attention [17].

On the other hand, electric actuators are evolving towards high-frequency and integrated design to increase power density. High-speed motors have distinct advantages in power density compared to traditional motors. As a result, they have been widely utilized. Additionally, wide-bandgap (WBG) power devices, represented by silicon carbide (SiC), present superior characteristics compared to conventional Si devices [18]. It has been proven that WBG devices have the potential to greatly enhance the power density and control performance of the system.

With the rapid progress of electrification and electric actuators, the MEA power system will inevitably encounter increasingly complex electromagnetic compatibility (EMC) challenges [4]. High-frequency switching actions of WBG power devices bring more serious electromagnetic interference (EMI). Firstly, high-frequency common-mode (CM) voltage can accelerate insulation aging of the motor, which can damage the motor insulation system and shorten the motor's service life [19]. Secondly, the presence of electromagnetic coupling can induce high-frequency voltage in the bearings and other mechanical and hydraulic devices of the actuator, increasing the risk of damage and maintenance costs [20]. Moreover, the controller faces crosstalk issues between the power circuit and signal circuit, which cause system malfunctions [21]. Lastly, the EMI of the electric actuation system can conduct along the power lines or radiate through confined spaces, interfering with the normal operation of other electronic devices.

In conclusion, the electric actuator is evolving towards higher frequencies and greater power density, which in turn brings about increasingly complex EMC issues. Therefore, predicting and suppressing EMI generated by motor drive systems is crucial for the EMC and electromagnetic safety [22]. There is a need to strike a balance between power density and EMI in order to meet EMC requirements [23]. This paper provides a comprehensive review of current research on EMC in motor drive systems. The objective is to offer guidance for the design of electric actuators. In Section 2, the challenge of the trade-off between high power density and EMC design is analyzed. Then, the EMI modeling and prediction methods are reviewed in Section 3. Section 4 summarizes the EMI suppression methods. Finally, Section 5 summarizes the state of art of EMC design and provides an outlook for electric actuators.

## 2. The Trade-Off between High Power Density and EMC Design

Whether it is an EHA or EMA, the motor drive system serves as the central power component. Optimizing the power density of the motor drive system is vital for the overall improvement in the power density of the electric actuator. To further enhance power density, two main areas of development are being pursued: the integrated motor drive (IMD) and the application of WBG power devices. However, striking a balance between high power density and EMC has become a challenging issue. Consequently, this section aims to discuss the current trends in the IMD and the impact of WBG power devices, while also highlighting the challenges in EMC design for electric actuators.

### 2.1. Integrated Electric Motor Drives

A traditional motor drive system connects the motor, power converter, and sensors through cables and signal lines. However, there are several drawbacks. Firstly, the use of cables increases the overall size and weight of the system, which consequently reduces the power density and efficiency. Additionally, cables can worsen EMI, which is undesirable. On the one hand, cables are the main propagation path for conducted EMI. Conducted EMI directly penetrates sensitive equipment through cables, or is coupled to adjacent cables or signal lines through crosstalk. On the other hand, cables are high-efficiency radiation antennas. Long cables will not only cause excessive radiated emissions, but also make the system susceptible to external radiated EMI, worsening electromagnetic sensitivity.

To suppress EMI problems caused by cables, EMI filters are usually used in commercial products and engineering applications, which unfortunately, reduce the power density. To suppress radiated EMI and improve the electromagnetic immunity of cables,

electromagnetic shielding solutions such as shielding cables and enclosures are usually used, which can also alleviate crosstalk problems. However, the shielding properties are not only affected by the electrical and geometric configuration of the shielding cable and enclosures [24,25], but also depend on the grounding schemes [26]. The design of effective electromagnetic shielding requires not only a theoretical foundation but also extensive engineering experience. Electromagnetic shielding also reduces power density. Considering the limited space of MEA and strict EMC standards, cables should be shortened as much as possible, and the IMD will become the dominant trend.

Through the integration, the converter and the controller housing take up less space, and no longer need long cables, which can suppress the EMI caused by cables and improve power density with 10%~20% less volume [27]. Reference [28] provides an overview of the current status of IMD systems and elucidates the opportunities and challenges they face. In the early stages of the IMD, the drive electronics are built into a separate enclosure that is mounted on the side of the motor, thereby reducing the length of the cable [29]. This partial integration has been commercialized in industrial applications, as shown in Figure 3a. However, the power density and volume optimization are still limited [30]. To further improve power density, the IMD is moving towards overall design and modular design, as shown in Figure 3b,c. Among them, the integrated modulated motor drive (IMMD) has received widespread attention [31]. The IMMD consists of multiple modular components, with each modular unit comprising a single motor stator pole and matching drive electronics, as shown in Figure 3c. The IMMD shows higher reliability, which is crucial for MEA [17].

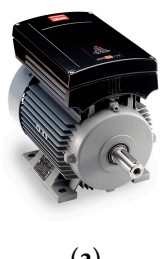

(**a**)

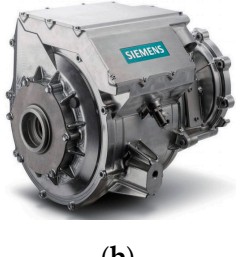

(**b**)

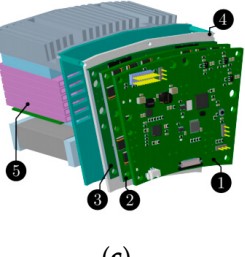

(**c**)

**Figure 3.** Examples of IMD: (**a**) Danfoss VLT FCM 300; (**b**) Siemens IMD technology for EV traction drives; (**c**) individual module of IMMD: (1) Communication board; (2) DC capacitor board; (3) Printed Circuit Board, PCB, of drive; (4) heatsink; (5) motor coils [32].

While the IMMD can indeed enhance power density and achieve fault-tolerant control of the actuator, it also faces several challenges, especially in heat dissipation and the EMI problem.

The heat in the IMD is generated not only from the copper loss and iron loss of the motor, but also from the switching loss and conduction loss of the power switching devices. The integrated design of the system makes it challenging to dissipate heat effectively, leading to overheating issues [33]. High temperatures can severely affect the performance of the system and may also cause damage to motor windings, switching devices, and drive circuits. The high-temperature working environment in the IMD has led to the application of WBG devices and posed higher requirements for system thermal management [34].

In addition, the IMD imposes stricter requirements for EMC design. EMC design aims to suppress EMI, with passive EMI filters being the most commonly used and most effective solution for reducing EMI. However, passive filters tend to be bulky and occupy up to 30% of the system volume, decreasing power density [35]. Moreover, the IMD puts forward new requirements for the trade-off between EMC and power density. Although, EMI can be reduced by filters, shortened cables and so on, the highly complex integrated design will bring new EMC challenges. Since the IMD controller is integrated inside the motor, the interference signal reflected by the motor housing can easily cause damage to sensitive

devices [28]. This presents a significant challenge to the reliability of electric actuators [36]. Unfortunately, there is still insufficient research on the reliability of sensitive devices.

### 2.2. Applications of Wide-Bandgap Semiconductors

In recent years, the application of WBG devices in MEA has emerged as a prominent research area due to the rapid advancement of WBG semiconductor packaging and integration technology [37]. As shown in Table 1 [38], compared to conventional Si devices, WBG devices such as SiC and GaN offer significant advantages in terms of blocking voltage capability, on-state loss, switching frequency, and high-temperature characteristics. The application of WBG devices effectively enhances the power density of electric actuators. The high switching frequency capability of WBG devices is suitable for high-speed motors with low inductance and high fundamental frequency [39], and also reduces the volume of filters and DC-link capacitors [40]. High frequency is the most effective method of increasing the power density [41]. Furthermore, compared with conventional Si devices, WBG devices exhibit less on-state loss and switching loss, resulting in reduced heat generation and improved efficiency. Consequently, WBG devices are well-suited for actuators in MEA, simplifying the heat dissipation design for IMD [28].

**Table 1.** Properties of WBG devices.

| Property | Si | GaN | SiC |
|---|---|---|---|
| Bandgap (eV) | 1.1 | 3.4 | 3.2 |
| Critical electric field (MV/cm) | 0.3 | 3.5 | 3 |
| Electron saturation velocity ($10^7$ cm/s) | 1 | 2.5 | 2.2 |
| Thermal conductivity (W/cm·°C) | 1.5 | 1.3 | 5 |
| Maximum operation temperature (°C) | 200 | 300 | 600 |

However, with WBG switching devices, the characteristics of noise sources are worse than Si devices, which leads to more severe conducted and radiated EMI. WBG devices exhibit higher switching speeds, higher switching frequencies, and more severe ringing than Si devices. Reference [42] investigates and quantifies the increase in the conducted CM EMI of motor drives with SiC and GaN devices. It is indicated that the influence of $dv/dt$ on the conducted CM emission is generally limited and the influence of switching frequency is more significant. Figure 4 illustrates the comparison of CM EMI between Si and WBG drives [42]. Due to the low parasitic capacitance and fast switch speed, the ringing or oscillation of WBG devices is much more severe. This can generate high-frequency EMI and seriously affect the reliability of electric actuators. References [43,44] investigate and model the mechanism of ringing and its impact on EMI.

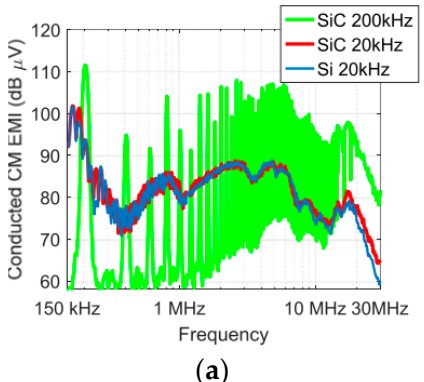

(**a**)

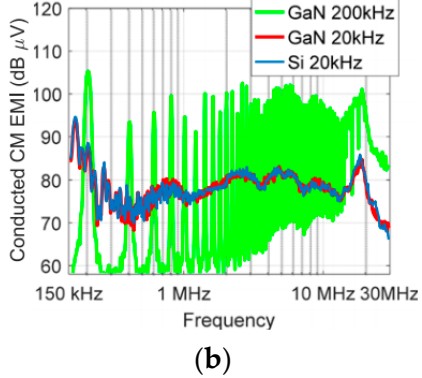

(**b**)

**Figure 4.** Comparison of CM EMI between the Si IGBT and WBG drives: (**a**) Si IGBT and SiC MOSFET; (**b**) Si MOSFET and GaN HEMT [42].

Although IMD technology and WBG technology can effectively improve the performance of electric actuators, it is essential to carefully consider the trade-off between power density and EMI and implement appropriate suppression strategies to meet EMC standards, such as DO160 or MIL-STD-461G. This paper aims to review the current state of EMI research on motor drive systems, including EMI modeling and suppression technology, to provide guidance for the design of high power density electric actuators.

## 3. EMI Modeling Methods

The purpose of EMI modeling is to reveal the mechanism of EMI generation and propagation, and to achieve accurate prediction, providing guidance for EMC design, thereby shortening the test time and cost. According to different propagation paths, EMI can be divided into conducted EMI and radiated EMI. The energy from conducted EMI is coupled with the power supply or other electrical equipment via the power line, while the energy from radiated EMI propagates through the electromagnetic field in space. The conducted EMI is defined with the frequency range from 150 kHz to 152 MHz, and the radiated EMI is from 100 MHz to 6 GHz in standard DO160. Although there is a lack of research on EMI modeling of electric actuators, the main objective of this paper is to provide a thorough overview of the research on EMI modeling of motor drives, thereby serving as a valuable reference.

### 3.1. Conducted EMI Modeling

At present, the conducted EMI modeling technology of a motor drive can mainly be divided into three types: the time-domain modeling method, the frequency-domain modeling method, and behavioral modeling.

### 3.1.1. Time-Domain Modeling

Time-domain modeling builds an equivalent circuit based on the physics model of the converter. The EMI spectrum is then obtained through time-domain simulation and frequency-domain calculation. Time-domain prediction relies on the accurate modeling of noise sources and propagation paths [45]. The core of noise source modeling lies in the precise description of the switching behavior of power devices. The equivalent circuit model is commonly used [46]. According to the datasheet provided by the manufacturer, the equivalent circuit can be obtained through software such as PSpice, Saber, and Simplore. The actual equivalent circuit needs to be validated and adjusted in combination with a double pulse test (DPT) [47]. To accurately model the propagation path, it is essential to consider the parasitic impedance of all components within the system [48]. As a result, broadband models of the line impedance stabilization network (LISN), passive components [49], power cables [50], and motor [51], and electromagnetic analysis of stray components generated in circuit layouts [52] should be investigated. Among them, the motor is the most critical and complicated. The motor is also typically modeled as an equivalent circuit model, which is more suitable for system-level simulation. Reference [51] proposed a behavior method based on series and parallel resonances in CM and DM impedance. The model considers the multiple resonances and low-impedance antiresonance. Therefore, the system-level EMI model has a high degree of accuracy, with a maximum prediction error of less than 5 dB across the frequency range of 100 MHz.

Although the equivalent circuit modeling can predict EMI over a wide frequency range, the electric actuator involves the coupling of electric, magnetic, and thermal multi-physics fields. The parasitic parameters are nonlinear, which affects the prediction accuracy. To address this issue, multi-physics co-simulation based on ANSYS is adopted in [53,54], which consider the time-varying, frequency-dependent, and thermal characteristics of key components.

Overall, time-domain modeling has the advantages of high accuracy, a clear physical meaning, strong portability, and ease of use. However, it should be noted that achieving

precise time-domain modeling can be time-consuming and requires high CPU performance, which can limit its practical application.

### 3.1.2. Frequency-Domain Modeling

In frequency-domain modeling, the EMI sources are simplified using approximations of equivalent voltage/current sources that mimic switching characteristics, while the propagation paths are still modeled as equivalent circuits, as with time-domain modeling. Ultimately, the EMI spectrum is obtained by calculating the spectra of the EMI source and the conduction path. By separately modeling and independently calculating the differential-mode (DM) noise and common-mode (CM) noise, noise separation can be achieved [55], as shown in Figure 5a. However, if the converter is asymmetric with respect to the ground, there is coupling between DM and CM noise, and the mixed-mode (MM) noise will limit the effectiveness of the noise separation [56]. Therefore, a differential–common-mode mixed model was proposed in [57], as shown in Figure 5b.

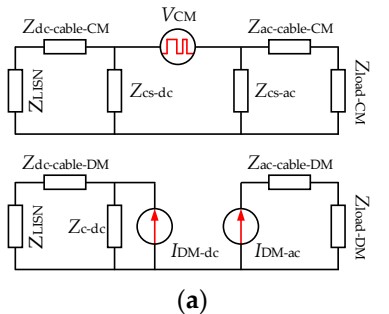 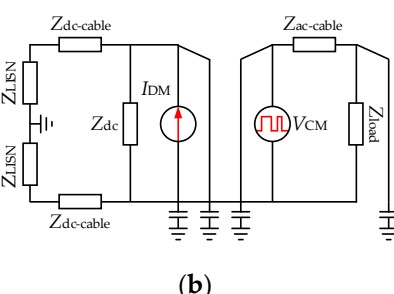

(**a**) (**b**)

**Figure 5.** Frequency-domain EMI models. (**a**) Noise separation model; (**b**) differential–common-mode mixed model.

The accuracy of the frequency-domain model depends on how well the noise sources are approximated. Typically, the noise sources are modeled as ideal trapezoidal waves with fixed slopes [55,58]. However, this ideal model fails to consider parameters such as switch junction capacitance, busbar lead inductance, and power module lead inductance. As a result, it cannot accurately predict nonideal switching characteristics, such as ringing. In [57], a refined noise source model based on simulation and experimental measurements is proposed, considering voltage overshoot, ringing, and reverse recovery effects. It can accurately predict EMI within 10 MHz. However, this method relies on experimental measurements. Reference [59] proposes a fast and precise synthesis strategy for noise sources, according to the switching characteristics of Si IGBT. It can quickly and precisely predict the ringing frequency, which can be used to achieve accurate EMI prediction. Compared to Si devices, the switching characteristics of WBG devices are more complex. Due to smaller parasitic capacitance, the ringing issues are more severe. Additionally, the Miller plateau time is shorter, resulting in a higher switching rate, and the reverse conduction characteristics are also different. As a result, the EMI peak frequency is higher and the amplitude is larger. Reference [60] introduces a method that considers both ringing, the Miller plateau, and reverse conduction by analyzing the transient behavior of WBG switches. It proposes a precise calculation method for the spectrum envelope of the noise source. However, this method has not been experimentally verified in complex motor drive systems.

In conclusion, compared to time-domain methods, frequency-domain methods have a faster calculation speed but lower accuracy.

### 3.1.3. Behavioral Modeling

Both time-domain modeling and frequency-domain modeling require detailed modeling of circuit components and consideration of all parasitic parameters. However, when it comes to complex motor drive systems, it is very difficult to accurately predict EMI using

detailed modeling [61]. To achieve a convenient and accurate prediction of EMI, behavioral modeling techniques have been developed [62]. Behavioral modeling treats the converter as a "black box" and uses a multi-port network (usually Thevenin or Norton circuit) to represent it. Through standardized measurements and numerical calculations, detailed parameters of the behavioral model can be obtained, enabling precise EMI prediction.

Behavioral models can be classified into terminated models and unterminated models. In terminal models, the load and inverter are represented as a single-port network. In [63], a three-terminal network is adopted to predict the EMI of the motor drive system, as shown in Figure 6a. The impact of the load on the noise source spectrum is also analyzed. This model can achieve precise EMI prediction within the frequency range of 30 MHz. The terminal model is relatively simple, but it is difficult to reflect changes in EMI caused by load variations. In unterminated modeling, the inverter is represented as a two-port network. Thus, the dc and ac side are independent, and the flexibility is strong. Reference [64] proposes a two-port network for the motor drive system to predict the CM EMI on the dc and ac sides, as shown in Figure 6b. For DM EMI, simplified single-port networks are used for predictions. This method accurately predicts EMI within the frequency range of 40 MHz. Reference [65] improved the extraction process of unterminated behavioral models, avoiding the influence of background noise during measurement on high-frequency EMI prediction. This method is suitable for motor drive systems that use WBG devices.

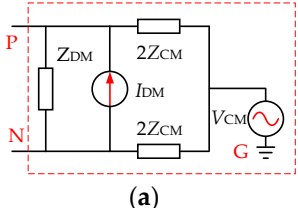

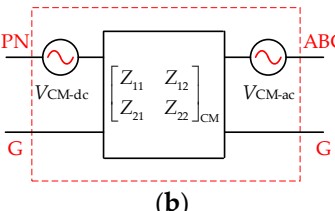

(**a**) (**b**)

**Figure 6.** Behavioral models for motor drive systems. (**a**) Three-terminated network [63]; (**b**) two-port unterminated network [64].

From a fundamental perspective, behavioral models approximate the nonlinear time-varying power electronic converters as linear systems [66]. In order to satisfy the assumption of linearity and time invariance, and to improve the prediction accuracy of behavioral models, the mask impedance must meet certain conditions, which poses challenges for the application of behavioral models [67,68]. In practical applications, MM noise also poses a challenge for filter design in noise separation [69]. In conclusion, the "black box" in the behavioral model does not have practical physical significance and cannot reveal the generation and propagation mechanism of EMI. However, due to its simplicity and high accuracy, it has been widely applied in the design of EMC filters.

### 3.2. Radiated EMI Modeling

Compared to conducted EMI, there has been less research on radiated EMI in motor drive systems. Motor drive systems primarily consist of motors, cables, and inverters, and the coupling mechanism of radiated EMI is complex. Numerical simulation methods [70,71] are usually used to calculate the radiated electromagnetic field. Figure 7 shows a typical implementation procedure for the radiated EMI prediction [71]. Further research is still required for the radiated EMI modeling of electric actuators.

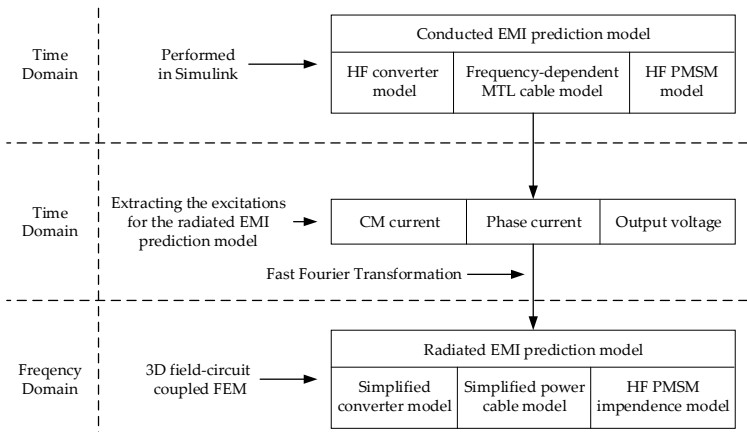

**Figure 7.** Implementation procedure for the radiated EMI prediction [71].

### 3.2.1. Radiated EMI Modeling of Cable

In a motor drive system, cables have a similar impact on radiation as high-efficiency antennas. They act as the primary source of radiated EMI. To reduce EMI, there is a current trend towards the IMD and thereby minimizing cable length. However, due to practical limitations in manufacturing and implementation, cables remain an integral part of motor drive systems in the current scenario.

The cable radiation modeling methods can be divided Into analytical methods based on multi-transmission line (MTL) theory [72] and numerical methods based on full-wave simulations. The numerical methods include the finite difference time domain (FDTD) [73], partial element equivalent circuit (PEEC) [74], and finite element method (FEM) [75]. Although numerical methods can evaluate and analyze the radiation effects of cables, the simulations are very time-consuming, especially for bent cables. Therefore, it is necessary to develop fast analytical modeling methods for cables.

In [76], the three-phase cable bundle is simplified to an equivalent single straight cable over the ground plane, as shown in Figure 8a. However, it is only applicable to ideal straight cables, without considering the tightly arranged and bent wiring of cables in engineering applications. Reference [77] improves upon the MTL theory by considering the impact of the proximity effect on charge distribution. The MTL matrixes are modified accordingly, and the Hertzian dipole method is then used to rapidly calculate the radiated electric field of arbitrarily bent cables, as shown in Figure 8b. This fast method offers high accuracy within the 1 GHz frequency range, comparable to numerical methods.

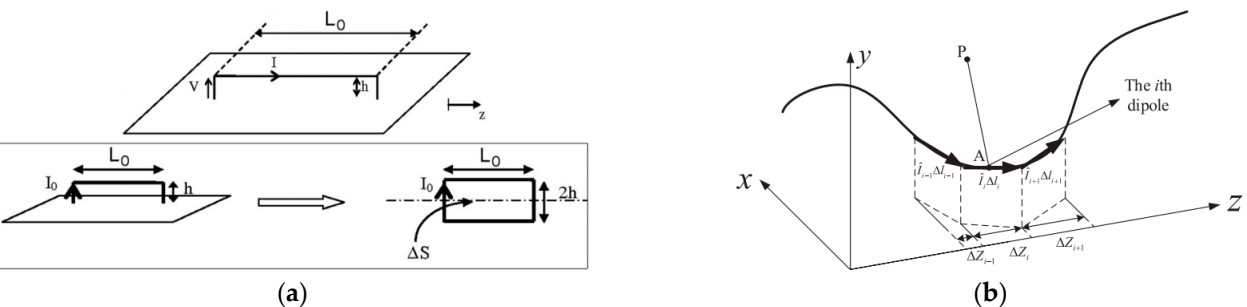

(**a**)　　　　　　　　　　　　　　　　　　　　　　　　(**b**)

**Figure 8.** Fast cable model for calculating the radiation. (**a**) Single straight cable model [76]; (**b**) arbitrarily bent cable model [77].

### 3.2.2. Radiated EMI Modeling of Motor

As an important component in motor drive systems, most studies explore the impact of impedance characteristics of the motor on conducted CM currents, as well as the indirect effects on radiated EMI [71,77]. In fact, the contributions of radiated emissions from the

motor itself are often neglected [78]. Engineering experiments have demonstrated that the use of high-power motors can lead to excessive radiated emissions. It is important to consider the impact of the structure of the motor, particularly the motor windings, on radiated EMI.

Currently, the motor radiation modeling method also involves analytical models or numerical methods. In [78], an analytical model based on a simplified winding structure is proposed, as shown in Figure 9a. The three-phase windings of the induction motor are equivalent to three Hertzian magnetic dipoles, and analytical expressions for the envelopes of the radiated emissions are deduced. Reference [70] develops a simplified numerical simulation method, where the three-phase windings are modeled as three 1D rectangular edge loops, as shown in the Figure 9b. Unfortunately, these simplified winding structures are only suitable for frequencies much lower than the resonant frequency of the motor. Based on a detailed winding structure as shown in Figure 9c, the radiation pattern of the stator winding is predicted using antenna array theory [79]. This can enable the prediction of the motor's radiation pattern before full-wave simulation. However, modeling a motor with many tightly packed winding wires in a complete full-wave model is challenging. To solve this problem, a reduction technique for modeling closely spaced wires was proposed in [80], which takes proximity effects into account. This approach considerably simplifies the full-wave model and accurately predicts the electric field radiation from 10 kHz to 30 MHz.

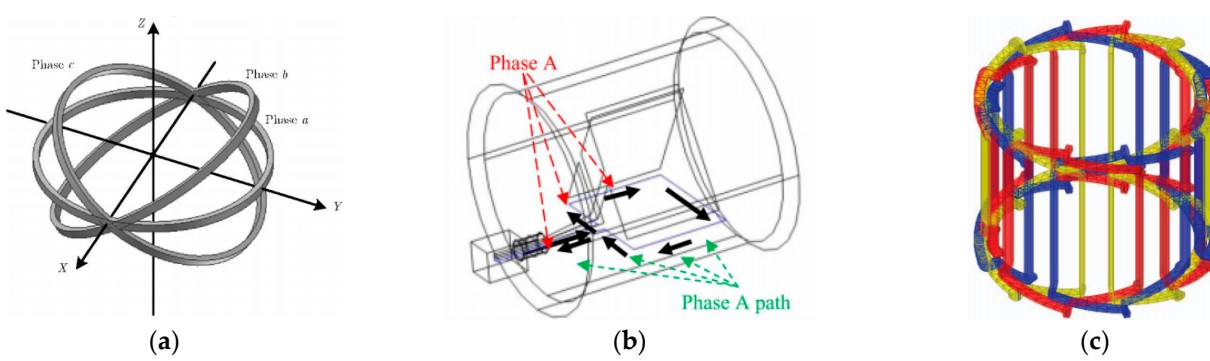

(**a**)  (**b**)  (**c**)

**Figure 9.** Winding structure of different radiation models. (**a**) Simplified winding structure based on Hertzian magnetic dipoles [78]; (**b**) simplified winding structure based on Hertzian magnetic dipoles [70]; (**c**) detailed winding structure [79].

### 3.2.3. Radiated EMI Modeling of Inverter

Radiation modeling of the inverter plays a crucial role in motor drive systems. The inverter, serving as the noise source, has a direct impact on the system's radiation, especially with the increasing applications of WBG devices. By acquiring the output signals of the inverter with different control parameters, utilizing the inverter port or circuit equivalent model in the radiated EMI model can facilitate a clearer and more intuitive investigation of the radiation mechanisms. It can also provide more targeted guidance for the suppression of radiated EMI.

A general radiated EMI model for a power converter is proposed in [81], as shown in the Figure 10. The input and output cables are treated as unintentional dipole antennas. The inverter is represented by a noise source and a source impedance in series. The current flowing through the antenna is the CM current of the system. The radiated electric field can be calculated using the developed model.

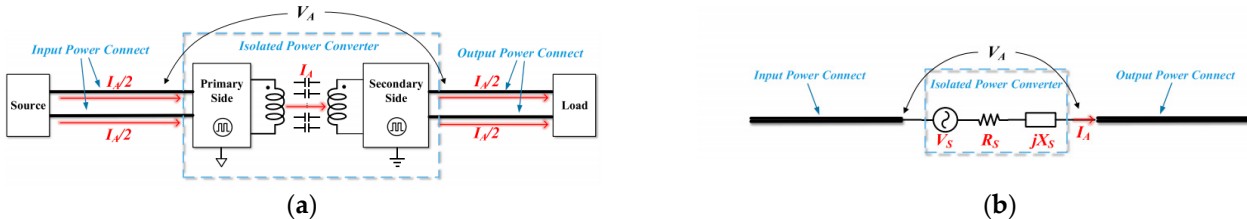

**Figure 10.** General radiated EMI model for power electronic converter [81]. (**a**) CM current flowing through the antenna. (**b**) Equivalent model.

In [82], the unintentional antenna is adopted to model the radiated EMI of the motor drive system with SiC devices. Time-domain analysis is adopted to obtain the accurate EMI spectrum of SiC devices. In addition, nonlinear parasitic capacitance, parasitic inductance, and load effects on the radiated EMI are taken into consideration. The predicted EMI results show good agreement with experimental results in the frequency range of 100 MHz to 1 GHz. With the applications of WBG devices, there has been a significant deterioration in radiated EMI within the frequency range of 150 kHz–30 MHz. At this moment, the radiation is exhibited as near-field coupling. According to Ampere's law, the total current in a cable is the sum of the conductive current and displacement current. However, in most studies, only the conductive current is considered, based on the assumption of far-field radiation. Reference [83] considers the coupling relationship between the displacement current and the antenna, effectively improving the prediction accuracy of radiated EMI in the low-frequency range.

## 4. EMI Suppression Method

The trend towards higher power density in electric actuators is expected to exacerbate EMI problems and worsen the overall electromagnetic environment of the system, presenting significant challenges for EMC design in MEA. EMI issues associated with electric actuators have become a major obstacle affecting the reliability of MEA. In this section, the state of art of EMI suppression methods for motor drive system is reviewed. As shown in Figure 11, EMI suppression methods are mainly divided into passive suppression and active suppression. Passive suppression utilizes additional passive devices or the optimization of parasitic parameters in circuits to achieve EMI suppression. Active suppression, on the other hand, suppress EMI by using active switch devices.

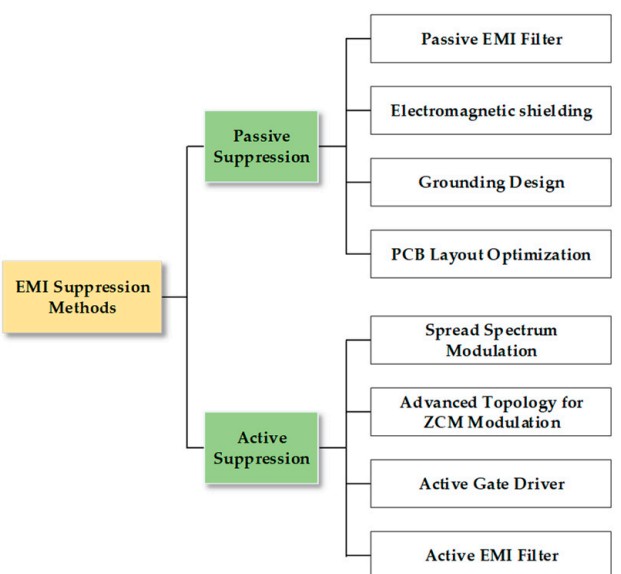

**Figure 11.** EMI suppression methods for motor drive systems.

### 4.1. Passive Suppression

The common passive suppression methods include four types: passive filter, electromagnetic shielding, grounding design, and PCB layout optimization. Grounding design serves as the basis of EMC, but relying solely on grounding makes it difficult to meet the EMI standards. Electromagnetic shielding [84] and PCB layout optimization [85] are the two main methods to suppress conducted and radiated EMI. However, these methods are primarily applied in DC/DC converters and are more intricate to implement in high power density motor drive systems. The passive filter is one of the most important and effective solutions for suppressing conducted and radiated EMI by improving propagation paths. The typical arrangement of a first-order CM/DM filter is shown in Figure 12.

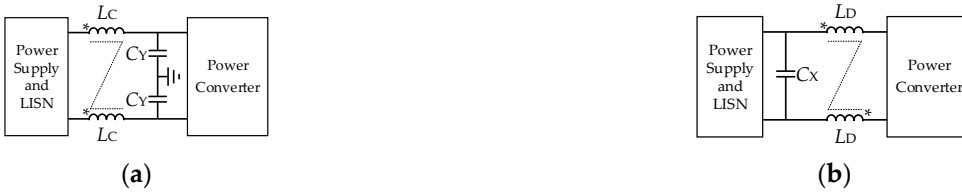

(a)                                                                   (b)

**Figure 12.** Basic structures of passive EMI filters. (**a**) CM filter; (**b**) DM filter.

In the common filter design, the EMI that exceeds the EMC standards is considered as the attenuation requirement or insertion loss requirement, without considering the influence of parasitic parameters. However, the insertion loss of passive filters is greatly affected by parasitic parameters. Figure 13 illustrates the equivalent circuit and insertion loss of a DM EMI filter considering these parasitic parameters [86]. It is evident from the figure that parasitic parameters significantly reduce the suppression performance of passive EMI filters in the high-frequency range above megahertz. In some cases, the presence of these parasitic parameters can even exacerbate the EMI in specific frequency ranges.

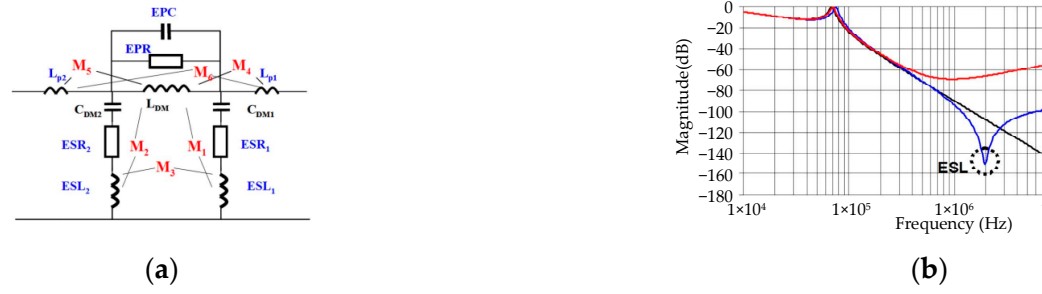

(a)                                                                   (b)

**Figure 13.** The influence of parasitic parameters on EMI filter suppression performance [86]. (**a**) Parasitic model of the passive filter; (**b**) effects of parasitic parameters on the insertion loss.

With the increasing application of WBG devices in power electronic systems, the switching frequency has greatly increased, resulting in a more serious EMI problem at high frequencies. The high-frequency performance of traditional passive EMI filters cannot meet the practical requirements. In addition, passive EMI filters are prone to being overdesigned to meet stringent EMC standards, and they tend to occupy a large volume in the system, which is not conducive to increasing power density.

#### 4.1.1. Optimization of High-Frequency Performance

To enhance the high-frequency performance of an EMI filter, it is crucial to account for the influence of parasitic parameters on insertion loss during the design procedure. Reference [87] analyzes the relationship between the filter structure's size and parasitic parameters, and proposes a design procedure with consideration of both the low-frequency and high-frequency attenuation requirements. Moreover, efforts should be made to minimize parasitic parameters. Reference [86] optimizes the structural layout, effectively

improving the self-parasitic and mutual parasitic problems. The insertion loss of the filters is significantly improved in the range of 1 MHz to 30 MHz.

In passive filters, near magnetic field emission from magnetic components is a crucial issue that can easily affect the high-frequency performance of the filter. In [88], a comprehensive analysis of magnetic coupling is proposed, considering the influence of the displacement current of parasitic capacitors on the stray flux. It reveals the frequency variation effect of a high-frequency magnetic field, and its influence on the performance of the EMI filter. The shielding plate is an effective solution to reduce magnetic coupling, but it is bulky and costly. As a result, a new topology of a CM inductor based on symmetric windings is proposed in [89]. It consists of two toroidal magnetic cores as shown in Figure 14. This topology reduces near-field emission due to DM currents and increases the DM inductance, at the expense of a reduced flux density. In the range above 2 MHz, the improved CM inductor reduces CM EMI by 10 dB compared to the conventional CM inductor, while DM EMI is reduced by up to 22 dB.

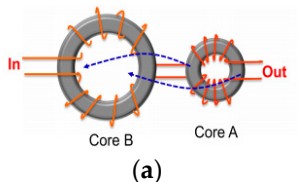

(**a**)

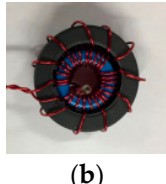

(**b**)

**Figure 14.** Novel CM inductor with reduced near-field coupling [89]. (**a**) Winding connection; (**b**) prototype.

For compact power electronic converters, the near-field coupling between the DC-link capacitors and the filter is also an important issue. Reference [90] studied the near-field coupling between the DC-link capacitor and the EMI filter in an active-clamp Flyback converter. It improves the single shielding by double shielding, as shown in Figure 15. Experiments have confirmed that double shielding effectively reduces CM conducted noise above 2 MHz. The double shielding technique can also be used to suppress radiated EMI [91].

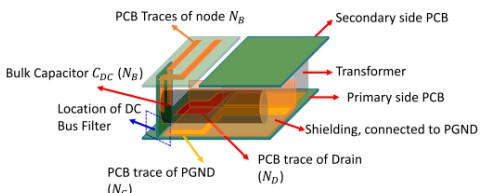

**Figure 15.** Double shielding technique to reduce near-field coupling [90].

4.1.2. Optimization of Power Density

Passive filters are one of the main limiting factors that hinder the improvement of a system's power density. Passive filter integration is a method for optimizing power density. It can be categorized into three levels: the functional level, material and technology level breakthroughs, and the system-level integration level.

Functional integration relies on magnetic integration technology. Figure 16b illustrates an integrated EMI inductor proposed in [92]. This design incorporates a low-permeability DM choke within the CM choke and improves the winding structure to achieve higher DM inductance. In [93], a DM solenoid is placed in the CM choke for magnetic integration, as shown in Figure 16c. This design allows for independent control of DM inductance, providing increased flexibility compared to the structure in [92]. In addition, reference [94] proposes a new method to realize the CM inductor on the AC and DC sides. Reference [95] realizes the integration of DM and CM mode inductors in a multi-stage filter.

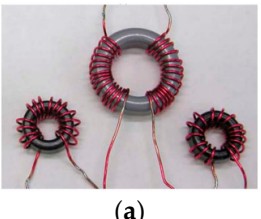

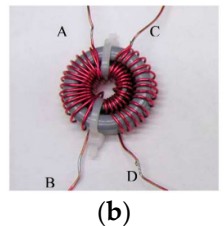

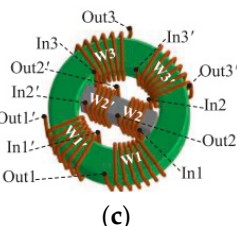

(**a**)  (**b**)  (**c**)

**Figure 16.** Structure of different EMI inductors. (**a**) Regular discrete EMI inductors; (**b**) integrated EMI inductor in [92]; (**c**) integrated EMI inductor in [93].

However, magnetic integration cannot realize the complete integration of capacitors and inductors, and the high-frequency performance of the filter is affected by the layout. To achieve a breakthrough in the power density of high-performance filters, advancements in materials and technology are required. Planar electromagnetic integration technology (EMIT) based on a PCB and flexible multi-layer foil (FMLF) has been developed. For instance, in [96], DM and CM filters are integrated using planar EMIT, and a ground layer is inserted to eliminate winding parasitic capacitance, enhancing power density and high-frequency performance. FMLF materials further reduce the loss and size of filters. Reference [97] proposes a fully integrated symmetrical filter based on FMIT, achieving a more compact design.

System-level integration enables the direct integration of the filter and power electronic converter. This is made possible through advanced packaging technology, which enables the integration of passive filters and power modules. Reference [98] integrates CM capacitors into SiC modules to reduce conducted EMI. Similarly, to minimize the filter parasitic effects, Reference [99] integrates the CM filter with the CaN half-bridge power module, as shown in Figure 17. Through this integration technology, up to 50 dB attenuation is achieved in the frequency range of 10 to 100 MHz. This modular integrated filter holds promising potential for application in electric actuators.

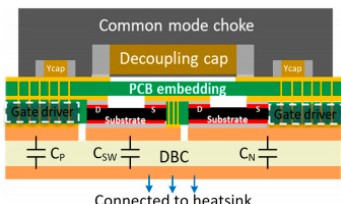

**Figure 17.** GaN half-bridge power module package with the integrated CM filter [99].

While passive EMI filters have seen advancements in high-frequency performance and power density, achieving a balance between power density, high-frequency performance, and manufacturing costs remains a challenge. The volume and weight of passive filters are still the main factors restricting the power density of an electric actuator.

### 4.2. Active Suppression

Active suppression technologies optimize EMI noise sources through strategies such as spread spectrum modulation, advanced topology, and an active gate driver, or actively detect and compensate EMI through active EMI filters. Compared with passive suppression, active suppression reduces the use of passive components and increases the power density of the system, so it has good application prospects in electric actuators.

#### 4.2.1. Spread Spectrum Modulation

The high-frequency switching of power devices controlled by PWM is the primary source of EMI in power converters. Traditional constant switching frequency PWM (CSF-PWM) concentrates energy mainly at the switching frequency and its harmonics, resulting

in EMI peaks at corresponding frequencies [100]. To reduce conducted EMI, spread spectrum modulation (SSM) is proposed by improving the noise source. Figure 18 illustrates that by adjusting the modulation, the narrow-band harmonic energy concentrated at the switching frequency and its multiples is dispersed over a wider spectrum range, leading to the attenuation of EMI peaks. Depending on the implementation, SSM is mainly divided into three categories: random PWM; periodic PWM; and programmed PWM.

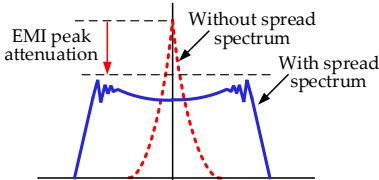

**Figure 18.** The principle of spread spectrum modulation [100].

Random PWM (RPWM) is a technique that utilizes the statistical properties of random numbers to modify the carrier characteristics in each switching cycle, thereby achieving the spread spectrum effect. RPWM is mainly divided into random carrier frequency PWM (RCFPWM) [101] and random pulse position PWM (RPPPWM) [102]. Reference [103] analyzes the impact of RCFPWM on the performance of power converters and verifies the effectiveness of RCFPWM in reducing conducted EMI. Reference [102] proposes an improved RPPPWM, which is easy to implement and integrate into motor drive systems. Then an improved RPWM that combines RCFPWM and RPPPWM is proposed, making the sampling frequency constant and harmonic cluster distribution more uniform [104]. RPWM has good performance in EMI suppression. However, the difficulty in generating random numbers hinders the practical application of RPWM.

In contrast to RCFPWM, periodic PWM changes the switching frequency periodically according to a certain law, such as sinusoidal wave, exponential wave, triangular wave [105,106], sawtooth pattern [107], or uniform distribution [108]. In [100], the influence of the peak deviation of the switching frequency on EMI suppression is analyzed. It is found that as the peak deviation increases, the EMI peak decreases. However, when the peak deviation reaches a certain level, harmonic overlap can occur, which counterproductively affects EMI suppression. Reference [108] analyzes the impact of the statistical distribution of switching frequency on EMI suppression, and proposes that the uniform distribution PWM improves the suppression effect. Since the switching frequency of periodic PWM changes periodically, inverter loss prediction can be performed to improve efficiency and optimize the thermal design [109].

RPWM and periodic PWM change the switching frequency with the goal of reducing the peak value of the EMI spectrum, which may have a negative impact on other performances of the motor and inverter. Programmed PWM, also known as model predictive PWM, uses the degree of freedom of the switching frequency to optimize specific indicators to improve the performance of the converter [110]. Variable switching frequency PWM (VSFPWM) uses the current ripple or torque ripple as the control object to change the switching frequency [111,112], achieving the multi-objective optimization of motor performances and EMI. Since EMI is not the primary target for programming PWM, the EMI suppression is unsatisfactory, compared with RPWM and periodic PWM.

### 4.2.2. Advanced Topology for ZCM Modulation

The CM voltage of the motor drive systems is determined by the switching actions of the inverter. However, the traditional three-phase two-level inverter topology lacks enough switching freedom, which prevents the improved modulation strategies [113] from maintaining a constant CM voltage. As a result, there still exist CM leakage of current issues. In order to eliminate CM current, several advanced topology solutions with multiple switching freedoms have been proposed, including three-phase four-leg inverters [114],

three-level inverters [115], paralleled inverters [116], and dual three-phase motors [117]. Schematics of these advanced topologies are shown in Figure 19.

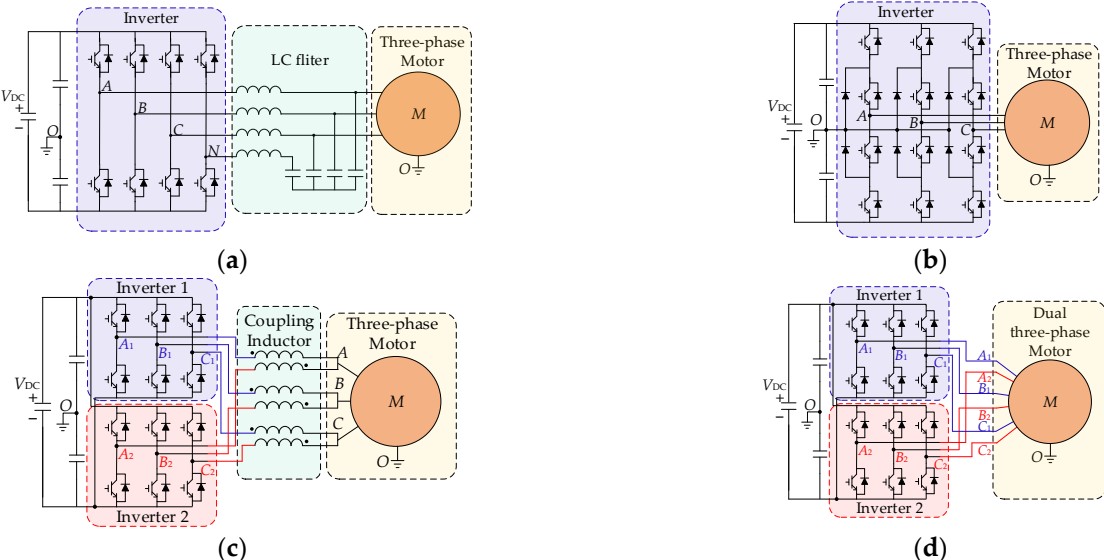

**Figure 19.** Schematics of advanced topologies for CM elimination. (**a**) Three-phase four-leg inverter; (**b**) three-level inverter; (**c**) paralleled inverter; (**d**) dual three-phase motor.

A zero common-mode (ZCM) PWM scheme, based on a four-leg inverter, is introduced for conventional three-phase motor drive systems in [114]. This modulation is based on the CM reduction modulation for a three-phase inverter [118]. By controlling the fourth bridge arm, the inverter output CM voltage is eliminated. However, the nonutilization of the zero vectors in the space vectors may lead to issues such as a reduced modulation index or increased total harmonic distortion (THD) of the output voltage.

Multi-level inverters have good application prospects in medium-voltage high-power motor drive systems [119], and they also have the switching freedom of the ZCM. A ZCM PWM scheme for a neutral-point-clamped (NPC) three-level inverter was proposed in [115]. ZCM modulation is achieved by selecting the switching vectors with a ZCM voltage state to synthesize the reference voltage. However, as with the four-leg topology, sacrificing part of the switching freedom to achieve ZCM leads to problems such as a deterioration in the output voltage, a reduction in the modulation index, and the unbalanced midpoint voltage of the DC-link. How to balance ZCM and other system performances is an important research direction.

Parallel inverters' topology is another topology used to achieve ZCM, which is suitable for high-power cases. By interleaving [120], the switching ripple between the two inverters, the THD, and output EMI can be reduced. However, interleaving cannot eliminate the CM voltage. A ZCM strategy for parallel inverters is proposed in [116]. The reference voltage is synthesized by paralleled ZCM vectors, and PWM timing optimization is performed to ensure the voltage balance and switching balance of the two inverters. This method not only achieves ZCM, but also reduces current ripple. In [121], a new dual-segment three-phase permanent magnet motor is introduced to eliminate the need for the coupling inductors. Without sacrificing switching freedom, the modulation index remains unchanged.

Due to the winding redundancy and reliability, modular winding motors such as dual three-phase motors have promising potential in electric actuators. Moreover, it is also beneficial to achieving ZCM modulation. A ZCM scheme is proposed for the dual three-phase motors with symmetrical windings in [117]. The output voltages of the two inverters are of the same magnitude but are opposite in phase. Moreover, a unified ZCM scheme is proposed for dual three-phase motors with asymmetrical windings in [122]. The universal method can achieve decoupling of the offset angle and reference voltage between the two

sets of windings. In [123], an interleaving together with ZCM modulation is proposed for a four-module three-phase motor to reduce the vibration and CM current at the same time.

Although advanced topologies with ZCM can eliminate CM voltage, it may also reduce the modulation index, increase switching loss, or increase ripple and harmonics. The trade-off between EMI suppression and system performances is important. In addition, the corresponding topology should be selected according to different application cases. Table 2 summarizes the reviewed publications on advanced topologies for ZCM modulation.

**Table 2.** Summary of advanced topologies for ZCM modulation.

| Topology | Ref. | Cases | Number of Power Devices | ZCM Performances | System Performances | |
|---|---|---|---|---|---|---|
| Three-phase four-leg inverter | [114] | Conventional three-phase motor drive | 8 full control switches | The CM voltage is reduced by 20 dB up to 100 kHz. | 1. | Reduced modulation index |
| | | | | | 2. | Increased THD |
| three-level inverter | [115] | Medium-voltage high-power motor drive | 8 full control switches and 6 clamping diodes | The CM voltage is almost up to 20 kHz. | 1. | Reduced modulation index |
| | | | | | 2. | Increased THD |
| | | | | | 3. | Unbalanced midpoint voltage |
| paralleled inverter | [116] | High-power motor drive | 12 full control switches | The CM voltage is reduced by 30 dB up to 200 kHz, and 10 dB up to 2 MHz. | 1. | Unchanged modulation index |
| | | | | | 2. | Improved current ripple |
| | [121] | | | The performance is the same as that in [121]. | 1. | Unchanged modulation index |
| | | | | | 2. | Improved current ripple |
| | | | | | 3. | Increased power density |
| dual three-phase motor | [117] | High-reliability motor drive | | The CM leakage current is reduced by almost 20 dB up to 40 kHz. | 1. | Unchanged modulation index |
| | | | | | 2. | Only suitable for symmetrical windings |
| | [122] | | | The CM voltage is reduced by more than 20 dB between 150 kHz and 900 kHz, and 10 dB up to 2 MHz. | 1. | Unchanged modulation index |
| | | | | | 2. | Suitable for asymmetrical windings |
| four-module three-phase motor | [123] | | 24 full control switches | The CM voltage is reduced by more than 30 dB between 150 kHz and 1 MHz, and 10 dB up to 3 MHz. | 1. | Unchanged modulation index |
| | | | | | 2. | Improved vibration |
| | | | | | 3. | Improved current ripple |
| | | | | | 4. | Only suitable for reversed windings |

### 4.2.3. Active Gate Driver

PWM determines the low-frequency spectrum of the noise source, and it has been proven that SSM and ZCM modulation can effectively suppress conducted EMI below several MHz. However, the high-frequency spectrum of the noise source is primarily determined by the switching transient of the power device, such as switching speed and ringing. The high-frequency conducted and radiated EMI can be reduced by increasing the gate resister (Rg) to slow down the switching speed [124]. Unfortunately, this method increases the switching loss [125]. To manage the trade-off between switching loss, device stress, and EMI, the active gate drive (AGD) technique has been developed. The AGD

adds active devices to the conventional drive circuit (CGD), and flexibly optimizes the switching trajectory.

To suppress high-frequency EMI problems caused by overcurrent and ringing, the AGD circuit in [126] controls the gate voltage according to different stages of the IGBT switching transient, reducing the switching speed. An integrated AGD is designed for GaN transistors in [127]. Based on closed-loop control, the drive current is reduced during the switching transient, reducing d$v$/d$t$ and switching losses. With this drive, the spectrum energy is effectively attenuated in the range of 30 MHz~200 MHz. In addition, a programmable AGD with sub-nanosecond resolution is designed for GaN transistors [128], which can accurately control the Rg and reduce the switching speed and oscillation. This method can reduce EMI noise in the range of 200 MHz~1 GHz. In fact, reducing the switching speed will still inevitably increase switching loss.

Usually the switching trajectory is a trapezoidal wave represented by the duty cycle and switching speed. It is proven that the higher the derivative order of the pulse wave, the faster its spectral attenuation speed [129]. The comparison of pulse waves with different derivative orders is indicated in Figure 20. Since the trapezoidal wave only has the one-order derivative (that is, the switching speed), its spectrum envelope decreases slowly. As a result, AGD methods based on Gaussian switching are proposed in [130,131]. The Gaussian S-shape has an infinite-order derivative, and since it does not change the switching speed, this method does not significantly increase switching loss. However, this method is difficult to implement and is not suitable for high switching frequency situations.

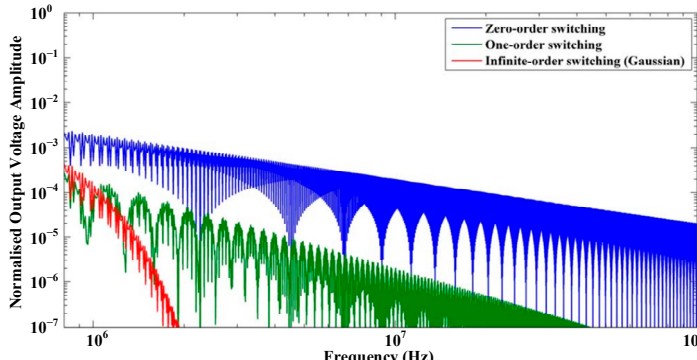

**Figure 20.** Comparison of pulse waves with different derivative orders: square wave with zero-order derivative (blue), trapezoidal wave with one-order derivative (green), Gaussian switching with infinite-order derivative (red) [130].

In summary, the AGD technique can effectively suppress both high-frequency conducted EMI and radiated EMI. The trade-off between switching loss and EMI is its main design basis. Currently, in the motor drive system, the application of AGD is insufficient and still needs to be explored and researched.

### 4.2.4. Active EMI Filter

Both the active EMI filter (AEF) and the passive EMI filter suppress EMI by reducing the effectiveness of the interference propagation path. However, AEFs actively sense the EMI and cancel it through integrated amplifiers, with a higher power density. The designs and implementations of different AEFs are summarized in [35]. According to the control method, AEFs can be divided into feedforward and feedback types. According to the signal types of sampling and compensation, they can be divided into voltage sampling voltage compensation (VSVC), voltage sampling current compensation (VSCC), current sampling current compensation (CSCC), and current sampling voltage compensation (CSVC).

The advantages of an AEF are reflected in two aspects. On the one hand, it is an effective way to suppress low-frequency EMI, which can increase the corner frequency of a passive filter and reduce the volume and weight of passive components. In [132],

a feedforward VSVC AEF, also known as an active CM canceller (ACC), is applied in motor drive systems to eliminate the CM current on the output side. In [133], an AEF based on CSCC topology is proposed for CM suppression of motor drive systems. On the other hand, an AEF can increase the equivalent impedance of passive components, thereby reducing their volume and weight [134]. In [135], a feedback VSVC AEF is proposed to increase the equivalent capacitance of the compensation capacitor, which improves the attenuation characteristics of the passive filter in the high-frequency range.

In an AEF, the transformer is required to implement current sampling or voltage compensation, which results in increased costs and reduced power density. Therefore, the application of transformerless AEFs has been widely studied. A novel transformerless AEF based on VSCC is proposed in [136]. A CM impedance network with the same CM impedance as the motor is built on the AC side, and the compensation voltage is injected into the network to generate compensation current. In addition, Reference [137] designs the AEF between the motor and the ground, which eliminates the transformer and reduces the current stress of the AEF, further optimizing the power density.

The application of WBG devices in electric actuators places higher requirements on the attenuation capability of the filter in a wide frequency range. However, due to the amplifier bandwidth, transformer bandwidth, and parasitic parameters, an AEF is more suitable for EMI suppression in the range of tens of kHz to several MHz. Therefore, the concept of a hybrid EMI filter (HEF) is proposed to improve the power density of systems [133,138]. Low-frequency EMI is suppressed by the AEF, and high-frequency EMI is suppressed by the passive filter, as shown in Figure 21.

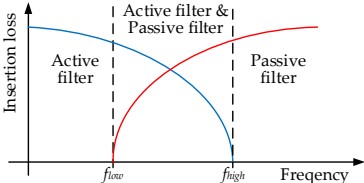

**Figure 21.** Insertion loss of the active and passive filters in a hybrid EMI filter [133].

## 5. Conclusions

This review introduces the current trends in electric actuator technology and the EMI problems it brings. The modeling and suppression methods for EMI in motor drive systems are summarized, providing guidance for the EMC design of electric actuators.

Due to the limited onboard space of MEA, high power density electric actuators have become the focus of research. As a result, electric actuators are developing towards integration and high frequency, with WBG devices being widely applied. However, the application of WBG semiconductors deteriorates the EMI of the system, posing higher requirements for EMC design, which in turn leads to an increase in the volume and weight of passive EMI filters, thereby reducing the power density of the system. Additionally, the complex electromagnetic environment of the IMD presents challenges to the reliability of sensitive devices such as micro-electronics components. Therefore, in future electric actuator designs, not only power density, efficiency, and reliability metrics must be considered but also the EMC.

EMI modeling and suppression are two important aspects of EMC design. This article provides a detailed review of the research on EMI in motor drive systems. EMI can be classified into conducted EMI and radiated EMI based on the different propagation paths. The research on modeling methods of conducted EMI is as follows:

(1)  The modeling methods for conducted EMI mainly include time-domain modeling, frequency-domain modeling, and behavioral modeling.

(2)  Time-domain modeling and frequency-domain modeling provide detailed modeling of converters, revealing the mechanism of EMI generation and propagation. They

provide a basis for the converter design. However, it is difficult to balance prediction accuracy, computational speed, and convergence.

(3) Behavioral modeling treats the converter as a "black box", which can achieve accurate and fast predictions. Behavioral modeling must rely on existing prototypes for modeling, meaning that it cannot provide guidance for EMC predesign. However, behavior modeling still plays a crucial role in guiding the design of filters.

The research on modeling methods of radiated EMI is as follows:

(1) Compared to conducted EMI, there is less research on radiated EMI modeling. Radiated EMI modeling can be mainly divided into three aspects: cable, motor, and inverter.

(2) With the development of IMDs, the impact of cables on radiation will gradually decrease, and the motor's radiation emissions will become the dominant factor in the future. Numerical methods are common modeling techniques, but they are time-consuming and computationally inefficient. Therefore, there is a growing research trend towards developing methods that balance accuracy and computational efficiency.

(3) Compared to the motor and cable, the inverter itself has lower levels of radiated EMI. However, as the noise source in motor drive systems, the research on the influence of the inverter on radiation is important.

The development of high power density electric actuators puts forward higher requirements for the trade-off between EMC and power density. At present, EMI suppression methods can be divided into two types: passive suppression and active suppression. The research on EMI suppression is as follows:

(1) Passive EMI filters are widely used and highly effective methods for suppressing electromagnetic interference (EMI). However, their bulky size poses a challenge to improving power density. Currently, research efforts are focused on optimizing the high-frequency performance and power density of these filters.

(2) Active suppression has gained increasing attention and research due to its advantages in high power density.

(3) SSM and ZCM modulation based on advanced topology are effective in suppressing electromagnetic interference for low-frequency conducted interference. However, using or sacrificing switch freedom to optimize EMI may lead to a deterioration in control performance, which is a crucial aspect to consider.

(4) AGD, which achieves noise attenuation by optimizing the switching trajectory, is effective in suppressing high-frequency EMI. However, compared to other suppression methods, AGD places higher demands on packaging technology and has limited applications in motor drive systems.

(5) Compared to passive EMI filters, AEFs can effectively increase power density. However, due to bandwidth limitations and parasitic parameters, AEFs are difficult to implement for effective EMI suppression across a wide frequency range. The combination of AEFs and passive EMI filters, known as hybrid filters, is currently a hot research direction.

**Author Contributions:** Conceptualization, Z.W. and D.J.; methodology, Z.W.; software, Z.W.; validation, Z.W., X.Z., Z.L. and G.Y.; formal analysis, Z.W.; investigation, Z.W.; resources, Z.W.; data curation, Z.W.; writing—original draft preparation, Z.W.and X.Z.; writing—review and editing, Z.W., G.Y. and H.L.; visualization, Z.W.; supervision, D.J. and Z.L.; project administration, D.J.; funding acquisition, D.J. All authors have read and agreed to the published version of the manuscript.

**Funding:** This research received no external funding.

**Data Availability Statement:** Not applicable.

**Conflicts of Interest:** The authors declare no conflict of interest.

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
