# Peer review of "A Review of EMI Research of High Power Density Motor Drive Systems for Electric Actuator"

_actuators, doi:10.3390/act12110411_

Round 1

Reviewer 1 Report

Comments and Suggestions for Authors

The paper addresses significant aspects of high-power-density motor drive systems, making it a compelling read. However, several areas require improvement prior to publication:

1. **Line 79, Page 3**: The authors introduce classical connections between motors and inverters via cables on this line. While the paper highlights the drawbacks in terms of size and weight, it's crucial to delve deeper into their impact on conducted and radiated EMI (Electromagnetic Interference). Additionally, exploring commonly adopted commercial solutions and using the classical approach as a reference for comparison would enrich the paper. Providing more information in this context is highly encouraged.

2. **Line 88, Page 3**: Please review the typesetting in this section. It appears that the word "imitated" should be replaced with "limited."

3. **Figure 19 Inverters**: The inverters presented in Figure 19 are intriguing, but the paper would benefit from a more comprehensive comparison. Consider including a detailed analysis of conducted EMI, either based on existing literature or simulations, to enhance the understanding of their electromagnetic performance. Furthermore, when comparing inverters b) and c), which employ the same number of active switches, it would be valuable to ascertain which of them offers superior power quality and EMI mitigation.

4. **General Improvement**: In general, the paper could be enhanced by incorporating more intricate details, thoughtful considerations, and supplementary information. This additional content would empower readers to make more informed decisions and better understand the nuances of high-power-density motor drive systems.

Comments on the Quality of English Language

The english is readable

Reviewer 2 Report

Comments and Suggestions for Authors

The submitted manuscript presents a review of the current trends in electric actuator technology and the EMI problems it brings. The modeling and suppression methods for EMI in motor drive systems are summarized, providing guidance for EMC design of electric actuators. The manuscript presents modeling and suppression methods both for conducted EMI and radiated EMI. The modeling methods of conducted EMI, including time-domain modeling, frequency-domain modeling and behavioral modeling is rather interesting. 

The manuscript is overall well written. The Authors clearly state in the introduction the objective of their research, they clearly explain the presented methodologies and innovations in the field of EMI suppression and the English level of the manuscript is high.

The only remark, would be to consider enhancing the proposed literature review. In the Introduction, the Authors could better explain what is the MEA concept and why it is important in modern aviation. This would give a better idea to the reader of why it is important to investigate EMI effects on MEA. Some suggested references are [Integrated supervised adaptive control for the more Electric Aircraft. Automatica 2020, 117, 108956] and [Stability and Control for Buck–Boost Converter for Aeronautic Power Management. Energies 2023, 16, 988] where innovative control techniques are proposed to control the power absorbed by actuators in the MEA framework.

Reviewer 3 Report

Comments and Suggestions for Authors

This paper reviews EMI-related topics on high-power density motor drive system. It states the actuator in aviation is actually a high power density motor drive system, especially in MEA(more electric aviation) and iMD(Integrated Motor drive) system.  

The review seems comprehensive, and appropriate including EMC modeling and suppression methods both in CE and RE as well. It also describes the trends in passive and active filters to be completed, and touches the converter topology to eliminate CM factors, illustrating the weak points of each method. I think this paper is well written, showing the state of art of EMC technology on the motor drive system. I thank the authors for their arduous, still fruitful job.

 I think this paper is well written, showing the state of the art of EMC technology on the motor drive system. I thank the authors for their arduous, still fruitful job. Following are some minor things and typos to be revised/corrected.

(1) At line 57:  "Firstly, high-frequency common-mode (CM) 58 voltage can accelerate insulation aging of the motor, which can damage the motor insulation system." This argument needs a reference to support your claim.

(2) At line 198:  The reference introduce => The reference introduces

(3) The characters in Figure 8, 10, and 143 need to be enlarged for better visual conditions.

(4) At line 339:  what do you mean by high-frequency static magnetic field? It would be better to be clear about this terminology.

(5) At libne 388:  4.1.1.  => 4.2.1

(6) At libne 421:  4.1.3.  => 4.2.2

(7) At libne 456:  4.1.3.  => 4.2.3

(8) At libne 484:  4.1.4.  => 4.2.4

Round 2

Reviewer 1 Report

Comments and Suggestions for Authors

Esteemed authors,

The revised manuscript has been diligently enhanced in accordance with the valuable feedback provided during the initial review round. The improvements made to the paper are commendable and have significantly strengthened its overall quality.

I am pleased to recommend the acceptance of this paper in its current form. The authors' dedication to addressing the previous concerns and refining their work has resulted in a manuscript that is now ready for publication.